# Mapping the dynamics of visual feature coding: Insights into perception and integration

**Tijl Grootswagers**[1,2]☯*, **Amanda K. Robinson**[3]☯, **Sophia M. Shatek**[4], **Thomas A. Carlson**[4]

**1** The MARCS Institute for Brain, Behaviour and Development, Western Sydney University, Sydney, Australia, **2** School of Computer, Data and Mathematical Sciences, Western Sydney University, Sydney, Australia, **3** Queensland Brain Institute, The University of Queensland, Brisbane, Australia, **4** School of Psychology, The University of Sydney, Sydney, Australia

☯ These authors contributed equally to this work.
\* t.grootswagers@westernsydney.edu.au

**Data Availability Statement:** Raw and preprocessed data are available online through openneuro: https://openneuro.org/datasets/ds004357. Supplementary Material and analysis

## Abstract

The basic computations performed in the human early visual cortex are the foundation for visual perception. While we know a lot about these computations, a key missing piece is how the coding of visual features relates to our perception of the environment. To investigate visual feature coding, interactions, and their relationship to human perception, we investigated neural responses and perceptual similarity judgements to a large set of visual stimuli that varied parametrically along four feature dimensions. We measured neural responses using electroencephalography (N = 16) to 256 grating stimuli that varied in orientation, spatial frequency, contrast, and colour. We then mapped the response profiles of the neural coding of each visual feature and their interactions, and related these to independently obtained behavioural judgements of stimulus similarity. The results confirmed fundamental principles of feature coding in the visual system, such that all four features were processed simultaneously but differed in their dynamics, and there was distinctive conjunction coding for different combinations of features in the neural responses. Importantly, modelling of the behaviour revealed that every stimulus feature contributed to perceptual judgements, despite the untargeted nature of the behavioural task. Further, the relationship between neural coding and behaviour was evident from initial processing stages, signifying that the fundamental features, not just their interactions, contribute to perception. This study highlights the importance of understanding how feature coding progresses through the visual hierarchy and the relationship between different stages of processing and perception.

## Author summary

This study investigates the foundational computations in the human early visual cortex that underlie visual perception. Using electroencephalography (EEG), neural responses to 256 grating stimuli with variations in orientation, spatial frequency, contrast, and colour were measured in 16 participants. We explored how the neural coding of these visual

scripts are available on github: https://github.com/Tijl/features-eeg.

**Funding:** This work was supported by an Australian Research Council (ARC) Discovery Early Career Researcher Awards awarded to TG (DE230100380) and AKR (DE200101159), and ARC Discovery Projects awarded to TAC (DP200101787). The funders had no role in study design, data collection and analysis, decision to publish, or preparation of the manuscript.

**Competing interests:** The authors have declared that no competing interests exist.

features and their interactions relate to perceptual similarity judgments. Results confirmed simultaneous processing of all four features with distinct dynamics, revealing conjunction coding for various feature combinations. Behavioural modelling demonstrated that each stimulus feature contributes to perceptual judgments. This work emphasizes the significance of understanding feature coding progression through the visual hierarchy and its relationship to perception from early processing stages.

## Introduction

The initial stages of human visual processing involve basic computations that form the foundation of our perception of the world. Neural populations within the primary visual cortex (V1) are responsible for encoding the basic features of a visual stimulus [1,2]. Within V1, neurons are selective for multiple features within the visual scene, including edge orientation, spatial frequency, contrast, and colour. Yet encoding of features is not confined to V1. Basic feature dimensions also have distinct responses in higher visual areas, for example colour coding in V4 [3]. Indeed, feature-specific coding has been extensively documented throughout the visual system [4,5]. However, there are still many open questions regarding how visual feature coding relates to perception. More recent work has investigated the link between low-level neural responses and perception, albeit in different contexts and not separating individual basic features. In one study, stimulus similarity judgements were found to be tightly related to early visual brain responses for geometric patterns [6]. Other work showed that low-level stimulus features had good concordance with neural responses, but were not predictive of perceptual judgements of scenes [7]. However, early visual responses and low-level models in these studies comprise multiple visual features that could have different contributions to perception. To really characterise the transformation of visual information across the hierarchy, we need to answer how these basic building blocks of vision (e.g., orientation, spatial frequency, contrast, colour) are combined and subsequently contribute to the overall percept of a stimulus. For example, which features have distinct effects on perception, and does the encoding strength of a feature predict its perceptual weighting? Understanding how features and feature combinations are coded across successive stages of processing can provide novel insight into the computations that transform visual input into perception.

Human neuroimaging methods with high temporal resolution have been increasingly applied to characterise the dynamic processes underlying visual feature coding in the brain. Orientation, spatial frequency, colour, and contrast information can separately be decoded from neural signals as early as 50 ms following a visual stimulus, and different features appear to have distinct temporal profiles [8–15]. These findings highlight the potential of time-resolved neuroimaging methods not only for tracking the coding of visual features in the brain, but also for revealing how coding changes over time. Temporal profiles are particularly informative in showing when feature coding is maximal, indicating the time that the most feature information can be extracted from the brain. However, many human neuroimaging studies focus on a single feature, making it difficult to directly compare temporal profiles across experiments with different methods. Therefore, a key missing piece of information is how multiple features are processed simultaneously, and how they interact.

High-level perception relies on combining features into a single percept. The binding problem refers to the challenge of combining information from separate neural populations from different modalities (e.g., sound and sight) or different features (e.g., colour and motion) into a unified representation. This problem may also apply within the same neural populations

[16]. For example, the observation that orientation and colour are both coded in V1 [17] does not imply that they are integrated together at this stage. Neural populations that code for multiple features could reflect an efficient feature coding-scheme and/or reflect coding of conjunctions (i.e., specific combinations of features). Studies using fMRI have shown plausible conjunction coding throughout the visual system, such as colour and motion [18], colour and form [19], and spatial frequency and form [20]. This work showed that the conjunction codes were different from the individual feature codes, but did not elucidate when the conjunction codes emerged. One possibility is that conjunction coding reflects feedback or recurrent processes, not early feedforward computations. If so, perhaps more time is needed to process conjunctions. Resolving these possibilities requires high temporal resolution population-level neural responses to reveal the timing of individual features compared to conjunctions.

In addition to understanding how individual visual features (and combinations of these features) are encoded in the brain, a critical aspect that has remained a challenge is understanding how encoding of individual features informs our perception of the stimuli. A recent development in the field has been to explore more deeply the extent to which neural coding informs conscious perception. High level information has been shown to be predictive of behaviour for object categorisation [e.g., 21–24] and scene perception [e.g., 7, 25–27]. It seems quite clear that neural responses to complex information such as form and category drive behaviour, but is the same true for the first stage of processing in our visual system? For example, are behaviourally relevant features coded more strongly in the brain? How fast do we get behaviourally-relevant coding? One possibility is that even the earliest coding of features is relevant for behaviour, but alternatively perhaps perception is more reliant on later processes.

To answer these questions, in this study, we aimed to map temporal response profiles, interactions, and behavioural relevance of basic visual features with different stimulus durations. We measured neural responses using electroencephalography (EEG) while participants viewed 256 oriented gratings that varied on four feature dimensions: orientation, spatial frequency, colour, and contrast. We used a highly efficient rapid presentation design which allows thousands of presentations in a short testing session. To assess to what extent feature processing, as an early visual process, is dependent on processing time, we used different presentation rates to target different depths of processing [28,29]. Previous work has shown slower presentation rates result in higher behavioural accuracies [e.g., 30] and stronger neural representations [29] than faster rates. Here, stimuli were presented in fast sequences at 6.67 Hz or 20 Hz, which bias to higher and lower stages of processing, respectively [29,31–33]. To link stimulus processing to perception, we collected independent behavioural judgements of perceptual similarity on these gratings (Fig 1). Taken together, this study contributes to three open questions: (1) What are the neural dynamics of basic feature dimensions in the human brain? (2) How do different features interact? (3) How do feature-related neural responses relate to observers' percepts? Our results reveal overlapping but distinct dynamics of the coding of features and conjunctions in the human brain, and demonstrate strong evidence that feature coding drives perception.

## Results

### Dynamic visual feature coding in the EEG signal

Our results showed highly overlapping dynamic information for orientation, spatial frequency, colour, and contrast (Fig 2A). Using multivariate pattern analyses to "decode" different levels of each feature (e.g., different orientations, different contrast levels), we observed featural information in response to images at both 6.67Hz and 20Hz. Group statistics of decoding onset (as indexed by first three consecutive time points BF>10) indicated that all four features

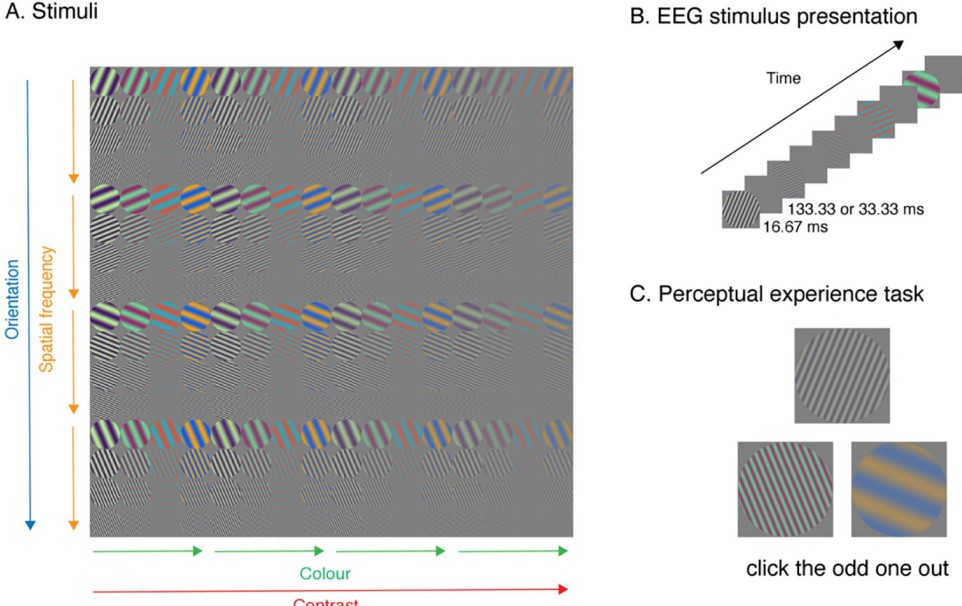

**Fig 1. Experimental design.** A) Stimuli were 256 oriented gratings that differed across four dimensions: spatial frequency, colour, orientation, and contrast. B) Example trials for EEG experiment. Stimuli were presented for one frame (16.67 ms) in sequences at 6.67 Hz (133ms ISI) or 20 Hz (33ms ISI). C) Example trials for the perceptual judgement task. Participants chose the stimulus that looked the most different (odd-one-out).

could be discriminated in the neural signal beginning from <90ms, at very early stages of processing (Fig 2A and Table 1). These times are consistent with early visual processing, likely beginning in the primary visual cortex. Supporting this reliance on early visual regions, sensor searchlight analyses showed central occipital sensors contained the most feature information not only at the peak of decoding (Fig 2A), but also for the second peak (see S1 and S2 Figs for searchlight maps over time). Notably, 95% confidence intervals of the onset times for the 6.67 Hz frequency were similar for spatial frequency, colour and contrast, indicating highly overlapping coding of these features in time. Orientation coding was lower overall, and that corresponded with a later onset of decoding, but it was still decodable in parallel with the other features. Contrast information peaked earliest, followed by spatial frequency and colour, and then orientation. Onset times were slightly later for colour and contrast in the 20Hz condition relative to the 6.67Hz condition, but peak times for decoding of each feature were nearly identical for the two presentation conditions, suggesting that feature processing was not affected by masking. Similar dynamics were observed for each individual (Fig 3), highlighting the reliability of these data. Time generalisation results revealed two pronounced stages of processing per feature (Fig 2B), with above-chance generalisation for training and testing on similar time points, and below-chance decoding for generalisation from one stage to the other (e.g., training on 100ms and testing on 180ms, and vice versa). These two-stage processing dynamics are consistent with previous work [34–36] and could reflect early stage feedforward and subsequent higher level recurrent processing [37]. Each feature exhibited slightly different patterns of generalisation suggesting the involvement of different neural populations (For full time-generalisation plots for both presentation frequencies, see S3 Fig). Together, these results indicate that orientation, spatial frequency, colour, and contrast have different coding dynamics, but that these processes operate in parallel, and the dynamics are highly conserved across different presentation conditions.

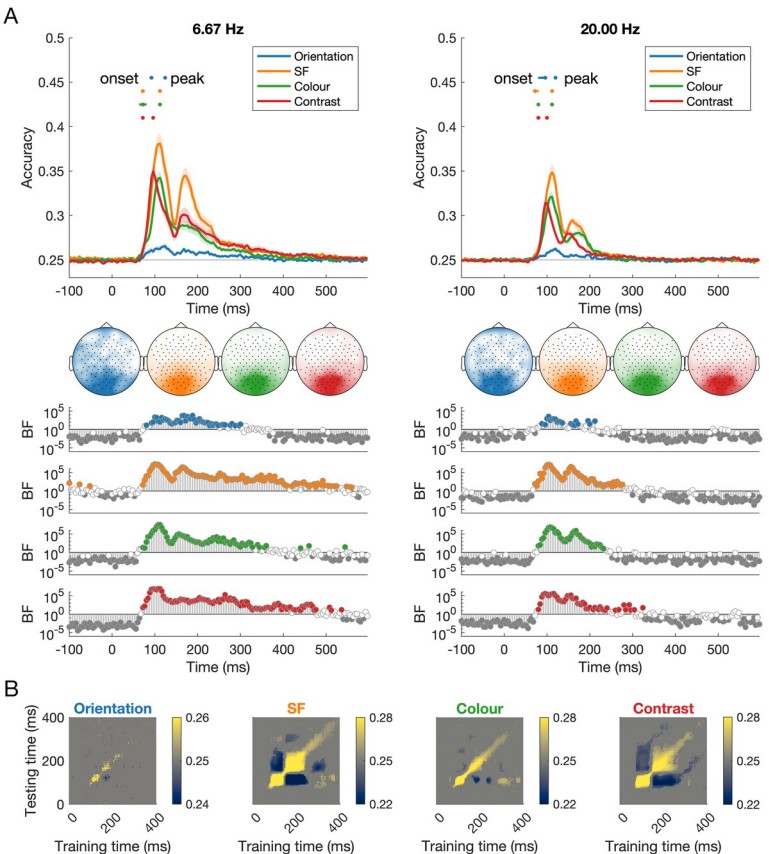

**Fig 2. Dynamics of visual coding for orientation, spatial frequency, colour, and contrast at different stimulus presentation rates.** A) The time course of decoding accuracy at 6.67Hz and 20 Hz presentation rates. Confidence intervals for onsets and peaks of individual features are plotted above the decoding traces. The head maps show the channel clusters with the highest feature information at the peak of decoding, based on results from a channel searchlight analysis (window size: 5 channels). Bayes Factors for classification evidence compared to chance (0.25) are plotted below. Coding peaked in order from contrast, then colour and spatial frequency, followed by orientation. The dynamics of feature coding were very similar regardless of presentation rate, though there was numerically higher classification accuracy for 6.67Hz compared to 20Hz. B) Time x time generalisation analyses for 6.67Hz condition show different above-chance and below-chance dynamics for each feature.

## Interactions and conjunctions of feature coding

Next, we were interested in how feature coding varied as a function of the stimulus content. We decoded orientation, spatial frequency, colour, and contrast for each level of the other features, for example decoding colour separately for each orientation (22.5, 67.5, 112.5 and 157.7 degrees). This analysis revealed interesting interactions that can be assessed by peak decoding accuracy (Fig 4). Notably, colour and contrast decoding decreased as spatial frequency decreased. Furthermore, spatial frequency and contrast decoding varied considerably depending on the colour of the stimuli, with higher decoding for the bluer/redder versus greener/purpler stimuli. Spatial frequency, but not colour, varied by contrast, such that the highest contrast resulted in the highest decoding accuracy. Classification of each feature did not vary consistently as a function of orientation, and orientation coding did not vary reliably across the levels of the other features. Note that although orientation coding was much lower overall, classification appeared above chance for each subset, showing that orientation information was reliably present in the neural responses despite its low signal (Fig 4, top row). Although

**Table 1. Timing of onset and peak of EEG decoding for features orientation, spatial frequency (SF), colour, and contrast, interactions between features, and for the correlation between EEG decoding and behaviour.** Cells with no values indicate no reliable onset time could be extracted. Temporal resolution was 4ms, and 95% confidence intervals are shown in square brackets.

| Feature | | Onset time (ms) | | Peak time (ms) | |
|---|---|---|---|---|---|
| | | 6.67Hz | 20Hz | 6.67Hz | 20Hz |
| Individual | Orientation | 92 [88–92] | 96 [80–100] | 124 [124–124] | 120 [120–120] |
| | SF | 72 [68–72] | 72 [68–80] | 112 [108–112] | 112 [112–112] |
| | Colour | 72 [64–80] | 80 [80–80] | 112 [108–112] | 112 [108–112] |
| | Contrast | 72 [72–72] | 80 [76–84] | 96 [96–96] | 100 [96–100] |
| Interactions | Colour x contrast | 88 [88–88] | 96 [96–96] | 148 [100–148] | 100 [100–100] |
| | SF x contrast | 84 [84–84] | 88 [88–88] | 172 [92–176] | 104 [104–104] |
| | SF x colour | 80 [80–80] | 96 [84–100] | 120 [120–124] | 124 [124–124] |
| | Orientation x contrast | - | 128 [88–164] | 100 [96–124] | 116 [116–328] |
| | Orientation x colour | - | - | 168 [168–168] | 108 [80–192] |
| | Orientation x SF | 104 [104–124] | 120 [100–120] | 136 [136–184] | 112 [112–120] |
| | Behaviour | 80 [80–84] | 84 [80–88] | 108 [108–108] | 112 [112–112] |

Fig 4 shows only 6.67Hz results, the same trend in results was seen for 20Hz presentation (see S4–S6 Figs for decoding traces at both presentation rates).

We used a different approach to investigate feature conjunction coding. The visual system is considered a hierarchy, beginning with independent visual feature processing (e.g., colour red, horizontal line) that ultimately seems to form a unitary percept (e.g., red line).

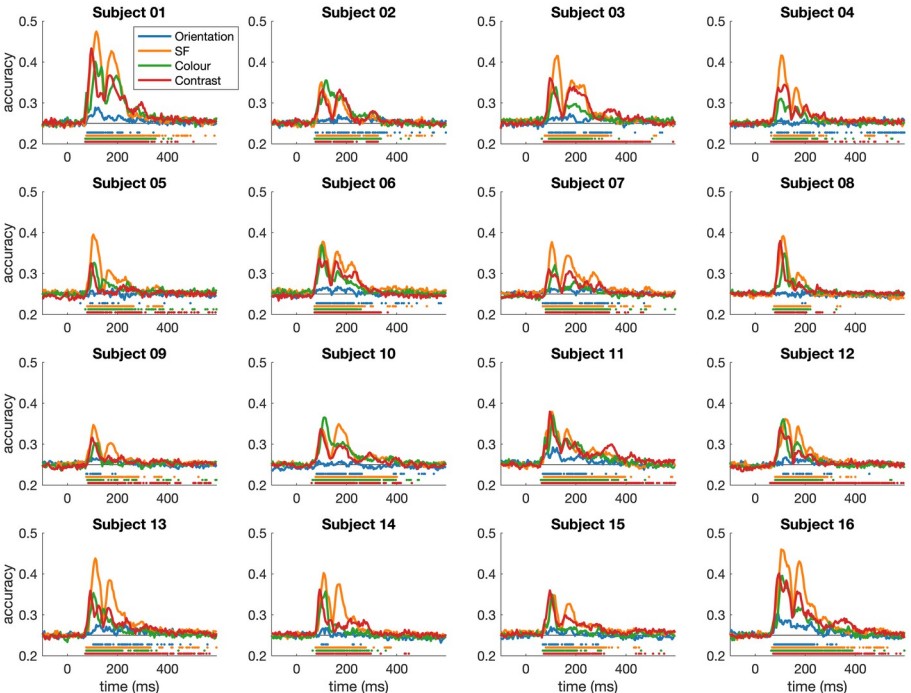

**Fig 3. Dynamics of visual feature coding for each individual subject.** Each panel shows the coding of orientation, spatial frequency, colour, and contrast for an individual subject. Marks above the x-axis indicate time points where decoding accuracy exceeded 95% of the pre-stimulus baseline accuracies. The individual plots show that feature coding observed at the group level (Fig 2A) is highly reliable and replicable at the individual subject level, with similar temporal dynamics overall.

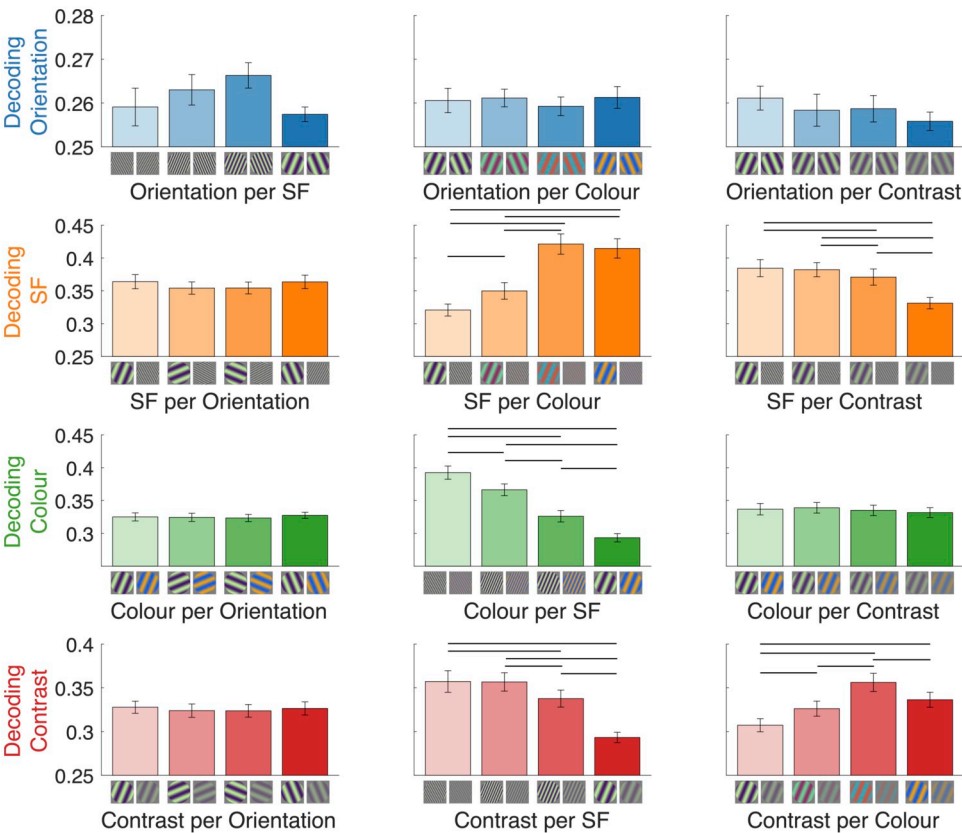

**Fig 4. Interactions between coding of different features at 6.67Hz.** Mean peak decoding accuracy for orientation, spatial frequency, colour and contrast (rows) according to each level of the other features (columns). Images below each bar depict two example stimuli that were decoded in that analysis. Lines above the bars mark conditions that differ reliably in decoding accuracy (BF > 10). Note the different y-axis scales per row. Y-axis starts at chance level, 0.25.

Understanding the scale and timing at which features become jointly processed can help shed light on how features are bound together in the visual system. To do this, we decoded groups of stimuli that each had the same absolute features, but varied in their feature *combinations*. We assessed conjunctions of two features at a time, using two levels of each feature in different combinations (see Fig 5A for the logic of this analysis). For example, to look at the conjunction of orientation and colour, one analysis classified between *orientation1/colour1* AND *orientation3/colour4* (Class A) VERSUS *orientation3/colour1* AND *orientation1/colour4* (Class B). Each analysis included all experimental stimuli with those features (e.g., across all spatial frequencies and colours), so there were 32 stimuli per class (Fig 5B). For each of the 6 feature combinations (colour × contrast, SF × contrast, SF × colour, orientation × contrast, orientation × colour, orientation × SF), there were 36 possible stimulus level combinations that were classified (e.g., 1/1 and 2/2 VS 1/2 and 2/1, 1/4 and 3/2 VS 1/2 and 3/4, etc). From these 36 analyses, the mean was calculated and compared to the chance level of 50% (see S7 and S8 Figs for results from each of the 36 analyses). Because the classes could not be distinguished based on any one feature alone, above chance classification for this analysis reflects unique neural responses to the *combinations* of different pairs of features, indicating the presence of feature binding.

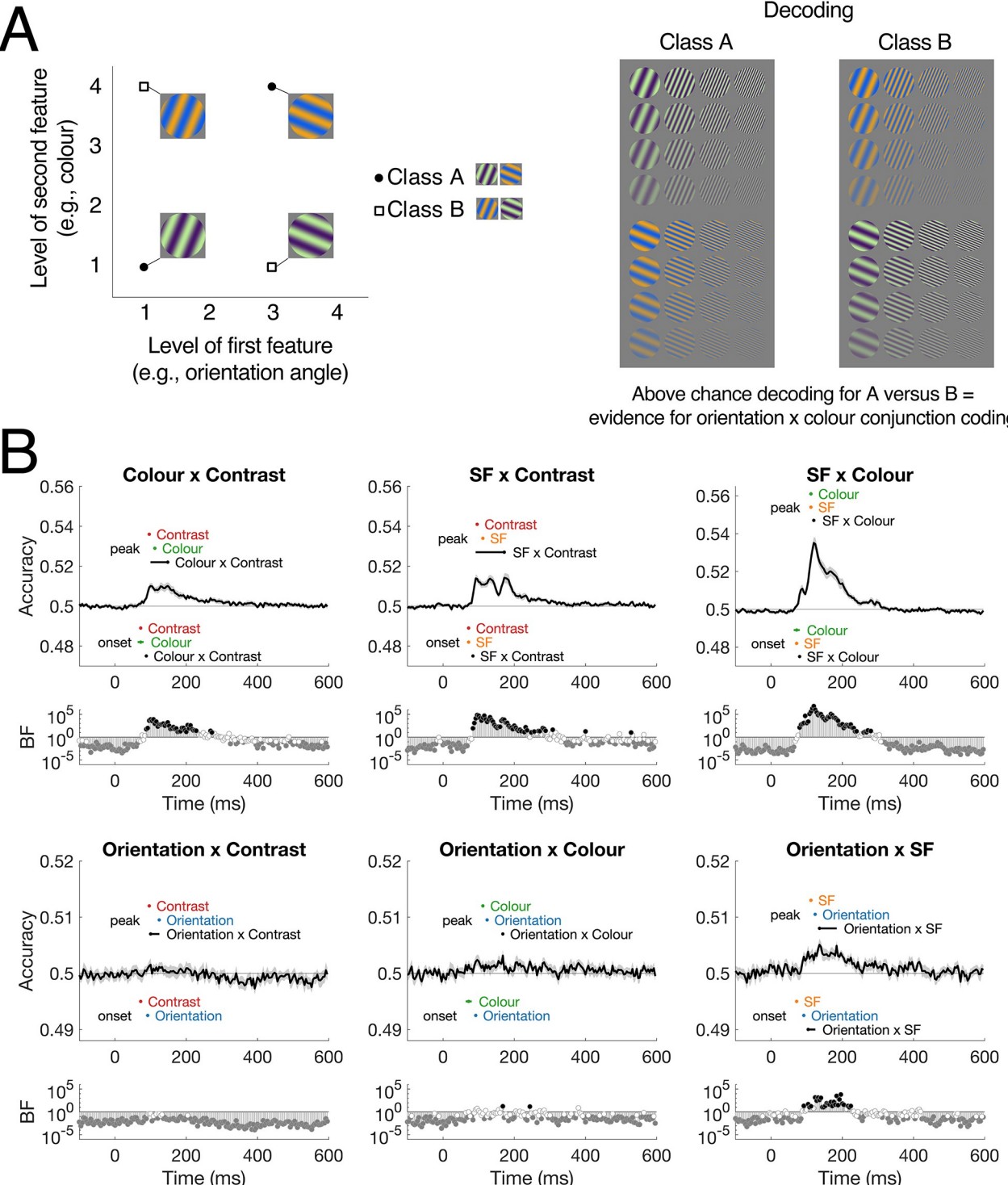

**Fig 5. Feature conjunction analyses.** A) Logic for the conjunction decoding analysis. Decoding classes came from two levels of two different features (e.g., contrast and orientation), so that each class could not be separated by differences in each feature alone. Classification was performed on groups of stimuli that together contained the same visual features (e.g., specific contrast and orientation) but varied on feature combinations. In this example, both classes contain two contrast levels and two orientations (and four colours and four SFs), so the only way to distinguish between them is by a neural response to the conjunction of contrast and orientation. B) Dynamics of feature conjunction coding for each feature combination. Onsets and peaks of individual features and conjunctions are plotted with 95% confidence intervals; onsets are below the chance level and peaks are above. Bayes Factors below each plot reflect the evidence for above-chance decoding; black circles are BF>10 and grey circles are BF<0.1. All results shown from 6.67Hz presentation rate. Note the different y-axis scales per row.

Fig 5B shows the dynamics of feature conjunction coding for each combination of features for the 6.67Hz presentation rate. These analyses showed reliable decoding for colour × contrast, spatial frequency × contrast, spatial frequency × colour and orientation × spatial frequency, reflecting specific neural processes for binding of these feature combinations. Decoding was reliable from early stages of processing, and different dynamics were evident for different feature combinations. Orientation × contrast, and orientation × colour did not show reliable above-chance decoding for any prolonged amount of time, though there were time points after 90ms for which there was uncertain evidence in support for the null or alternate hypotheses, indicating a lack of power for these analyses. There was a reliable progression in interaction coding onset, with the earliest coding for SF × colour, then SF × contrast then colour × contrast and finally orientation × SF. Notably, these onset times were later than coding of their respective individual features, indicating that feature binding was a subsequent stage of processing after individual feature coding, at least 8ms later. Though the peak times confidence intervals were quite large for some conjunctions, it was particularly evident for colour × SF that the peak occurred at least 8ms later than the peak of information for each feature alone, providing further evidence of distinct neural populations coding for the conjunction of features compared with individual featural coding.

## Representations of visual features and their relationship to behaviour

In a final set of analyses, we assessed how specific visual features and their associated neural codes related to behaviour. To assess perceptual similarity of the stimuli, a separate group of 530 participants completed 500 trials in an online experiment. In each trial, participants were shown three stimuli and were asked to choose the stimulus that looks the most different (i.e., the odd-one out; Fig 1C), and across 265,000 participant trials of various stimulus combinations, we quantified perceptual similarity of stimulus pairs. We used the Representational Similarity Analysis (RSA) framework [38] to compare neural dissimilarity across different stimuli with dissimilarities computed from visual feature models and behaviour. Neural dissimilarity matrices were constructed using pairwise decoding for all pair combinations of the stimuli, resulting in a symmetrical 256 × 256 matrix for each time point (32640 unique values). Analogous models were constructed for each of the four visual features (contrast, colour, spatial frequency, and orientation) based on the rank similarity of that feature in the stimulus set; importantly, these feature models were orthogonal, with no reliable correlations between them (Fig 6A). Behavioural dissimilarity matrices were constructed from the results of the odd-one-out task (Fig 1C), where higher dissimilarity indicated pairs of stimuli where one of the stimuli tended to be chosen as the odd-one-out when they were presented in triplets together. Fig 6B shows a 2-dimensional embedding of the perceptual dissimilarity between stimuli.

We observed interesting relationships between the neural, feature, and behavioural models. First, we assessed whether visual feature differences were reflected in behaviour (Fig 6A). There were particularly high correlations between behaviour and models of spatial frequency ($r = .567$, $p < .001$) and colour ($r = .497$, $p < .001$). Lower, but still reliable correlations were observed between behaviour and contrast, ($r = .118$, $p < .001$), and behaviour and orientation, ($r = .054$, $p < .001$), indicating that all four feature dimensions were likely being used to assess perceptual similarity of the stimuli. Thus, stimuli that were more similar in terms of spatial frequency, colour, contrast, and orientation tended to be perceived as more similar overall. We then assessed how the neural models related to the feature models and behaviour (Fig 6C). There were high correlations between neural and perceptual similarity from very early stages of processing (onset = 80ms), indicating that early visual responses, rather than purely high-level representations, were relevant for behaviour. The same trend was observed for the 6.67

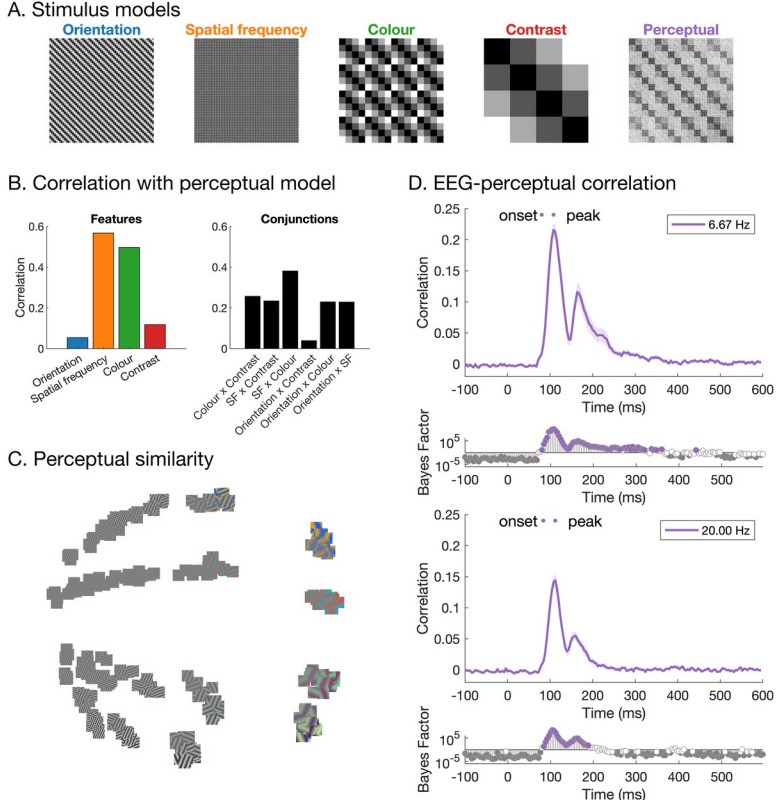

**Fig 6. Representational similarity analysis.** A) Models (representational dissimilarity matrices) based on stimulus orientation, spatial frequency, colour and contrast, and the perceptual model which was based on perceptual similarity judgements. B) Correlations between the feature models (and their conjunctions) and the perceptual similarity model. C) Similarity between stimuli based on perceptual similarity judgements, plotted using multi-dimensional scaling. The distance between stimuli reflects perceptual similarity, where images that are closer together tended to be judged as more similar perceptually. D) Correlations between neural RDMs and the behavioural model for 6.67Hz and 20Hz presentation rates. From early stages of processing, there are high correlations with the perceptual similarity model.

Hz and 20 Hz presentation rates. Importantly, the neural-perceptual correlation remained for hundreds of milliseconds, suggesting that prolonged neural feature representations are useful and important for perception.

## Discussion

Even the most complex processes underlying visual recognition rely on computations of basic visual features such as orientation and spatial frequency. Yet, we still know very little about interactions between features, and their relationship to behaviour. In this study, we investigated how visual features are coded in the brain using a large stimulus set designed to vary on features of orientation, spatial frequency, colour and contrast values. With fast image presentation, EEG, and multivariate decoding, we found that these features were processed in parallel in the brain, but varied in the strength and time of maximum coding. Stimulus presentation speed was varied to differentially limit visual processing [29], which showed that image masking influenced the strength but not dynamics (e.g., peak times) of feature information, indicating a particularly robust mechanism for visual feature coding. Further, we show that two-feature interactions and conjunctions are not all coded equally, implying separable neuronal populations are involved for interactions between different features. Behavioural similarity judgements of these stimuli

relied on all four features and correlated with neural information from early stages of processing, yet was sustained for hundreds of milliseconds. Together, these results indicate that the earliest visual feature processes contribute to perception, but that prolonged representations of an image's constituent features are relevant and important for robust perception.

## The time course of multiple simple features

Information about orientation, spatial frequency, colour, and contrast was represented in parallel, but with different dynamics, indicating that at least partially separable neural populations are responsible for coding these features. Our results were consistent with previous work, for example showing colour [11,13,14], spatial frequency [12], and orientation information [e.g., 39,40] can be decoded from high temporal resolution neural signals in humans. Going beyond previous studies, we were able to show how multiple features of the same stimulus are coded simultaneously and sustained for prolonged periods. This prolonged coding of each feature support prior research showing feature coding in multiple areas of the visual pathway beyond V1 [e.g., 9,18,41–43]. We found distinct representations for each feature, reflecting separable neural populations and/or different codes of multiplexing cells specialised for coding each feature. Time of peak coding, in particular, indicates the stage of processing at which the highest amount of feature-specific information is coded, shedding light on the representations of different features in the visual system. We acknowledge that absolute timing estimates can vary across studies due to specific laboratory and analysis details, so we focus on the differences in timing between conditions. Coding of contrast peaked earlier than the other features, supporting its role as a modulator of other feature coding [44], whereas spatial frequency and colour peaked slightly later, and orientation coding was later again. This is consistent with previous work showing later decoding for orientation than spatial frequency [12].

Coding of each feature exhibited a pronounced double-peak response, with the first peak around 90-130ms and a smaller second peak around 160-200ms, which corresponded with two distinct stages in the time generalisation analyses. These times are associated with the P100 and N1/N170 event-related potential components, which are typically described as two different stages of visual processing, likely reflecting low-level versus high-level (i.e., face/word/object) image discrimination, respectively [45,46]. Previous studies using EEG decoding have shown a multi-peak curve for orientation [39] and object decoding [29,32,36], though interestingly decoding curves from magnetoencephalography (MEG) data do not seem to show pronounced second peaks for orientation [40], spatial frequency [12], colour [11,14,15] or object information [34,47,48]. This points to the second peak in EEG reflecting neural responses from radial dipoles that are difficult to measure with MEG. One potential explanation is that the second peak reflects feedback or recurrent processes. Alpha oscillations have been implicated in feedback-related coherent visual processing [49] and notably, frontal alpha and theta oscillations are difficult to measure using MEG [50]. Frontal oscillatory activity might therefore play a role in high level visual representations after 150ms, in a potential feedback process that influences processing within occipital cortex. Supporting this idea, the searchlight analyses showed the most feature information was constrained to central occipital electrodes for both the first and second peak of decoding (see S1 Fig). Together, the two-stage process for feature distinctions we observed might therefore indicate two sweeps of information; one that is primary feedforward, and one that is primarily feedback [51]. This could play into why we (and others) found below-chance classification accuracy for generalisation between different stages of processing: a reversal of information flow might correspond with a reversal of neural information patterns such that images that have stronger feedforward representations might result in weaker feedback information.

### Early visual cortex responses are robust to changes in presentation rate

Neural dynamics were highly replicable across two different presentation rates (6.67Hz and 20Hz, with the same image duration), showing feature coding is reliable even with limited processing time. This is not surprising given that feature coding relies on early visual cortex which has rapid temporal tuning, and temporal resolution decreases across the visual hierarchy [9,52]. While we observed small differences for onset times and peak strengths between the 20Hz and 6.67Hz conditions, we do not consider this strong evidence for delays in neural coding because these measures can be influenced by overall signal-to-noise ratio [53,54]. Therefore, given the same peak times and overall dynamics, we consider feature coding unaffected by presentation rate. This is in stark contrast to previous work that used RSVP and decoding, which reported strong effects of presentation rates on decoding accuracies for object stimuli [29,32,33,39,55]. This is likely because the features investigated in this study predominantly rely on feedforward coding in the early visual cortex, whereas the object identities decoded in previous studies rely on the ventral temporal cortex which requires more recurrent processing and is therefore more affected by the masking in fast RSVP. Together, these results demonstrate that feedforward neural processing is highly robust to masking.

### Prolonged feature coding and cascading processes in early visual regions

Despite the likely reliance on early visual regions, stimulus feature coding was evident for prolonged periods of time. We found, for example, that spatial frequency and contrast could be decoded from ~70–500 ms in the 6.67Hz condition and from ~100–300 in the 20Hz condition, even though stimuli were only presented for one frame, 16.66 ms, and were irrelevant to the fixation detection task. This prolonged coding has two implications: cascading processes and temporal overlap. The current study replicated the temporal overlap observed in previous studies, where successive events can be decoded even though their neural processing signals overlap in time [32,56]. Here, we found that information about approximately three stimuli could be extracted from the neural signal simultaneously. Importantly, however, simultaneously processed stimuli seem to be held at different processing stages, which highlights potential mechanisms for the brain to accommodate for delays in processing and to keep track of time [cf. 57]. The other implication of prolonged information coding is that initial neural responses in early visual areas trigger a robust processing cascade. Despite very brief image presentations, characteristic coding dynamics were observed for each feature, involving at least two distinct stages of processing. The feature-specific nature of this coding is also interesting: rather than initial feature coding that transforms into a more feature-invariant "object" code, later processes still contain feature-specific information about multiple visual features. This suggests that high-level processing relies on maintaining feature information through recurrence.

### Relatively weak decoding of orientation

Oriented gratings have for decades been reliably and strongly decoded from fMRI data[42]. It is notable that in our study, orientation decoding was relatively low (albeit reliable) compared to other visual features. Previous work similarly reported low orientation decoding accuracy in EEG [e.g., 39], but higher in MEG [8,40]. All these studies used gratings with randomised phase, as we did, so variations in phase cannot explain our low orientation decoding accuracies. Furthermore, our design was highly powered, with 40 repetitions of all 256 stimuli resulting in 10240 events for each presentation rate and 2560 repetitions per orientation angle. Notably, we did not include cardinal or oblique orientations in our stimulus set, which likely reduced our power in detecting differences between orientations [40]. It could be that the

neural populations distinguishing between colour, contrast, and SF are more distributed, with asymmetries in their absolute signal strength (e.g., high contrast drives stronger responses [58]), which would make their respective sources easier to distinguish at the scalp. However, robust object perception needs to be invariant to rotation to some extent [36,59]. Orientation differences could be interpreted as a rotation of the same object, which would reduce its importance in our perceptual similarity task. While local orientation detection is crucial to find edges and shapes, at a higher-level orientation may not be as important, which could account for its weaker decoding compared to other features.

## Reliable coding of feature interactions

Feature interactions can shed light on the overlapping nature of neural populations coding for individual features. We assessed these interactions, by looking at how feature coding varied as a function of the other features. We found several interesting interactions, for example that colour coding decreased as spatial frequency decreased, and vice versa: spatial frequency coding changed across the experimental colours. In light of behavioural results showing that humans tend to perceive mixtures of colours at high spatial frequencies but separable colours at low spatial frequencies [60,61], these results point to higher overlap in neural populations coding for colour and high spatial frequency compared with low spatial frequency information. We also found a decrease in contrast coding with decreasing spatial frequencies, and complementary decrease in spatial frequency coding with decreasing contrast. These results fit well with the larger history of perceptual interactions documented in the psychophysics literature focused on contrast sensitivity functions, where spatial frequency sensitivity varies across contrast [62–64]. Recent work has found evidence that red/green gratings have higher contrast sensitivity than yellow/violet, which is supported by our findings that spatial frequency and contrast were more highly decodable for the bluer/redder gratings than the green/purple gratings [65,66]. Interactions for the other features have been documented but have not been studied extensively [65,67]. We found no reliable interactions between orientation and other features, possibly because orientation decoding was low in general, though it was consistently above chance. Notably, interactions were not exclusively bidirectional. So, for example, contrast coding varied across colour, but colour decoding was not influenced by contrast, a result that indicates a difference in the modulatory role of each feature. These interaction results were highly reliable across the two presentation rates. The interactions we observed point to partially overlapping neural populations responsible for coding different features.

## Separable neural processes for conjunction coding

The brain needs to bind different types of features to form a single percept, yet the neural mechanisms underlying feature conjunctions have remained elusive. In an effort to understand how feature conjunctions are coded, we implemented a novel set of analyses to extract information about conjunctions of two features from the neural signal, by comparing sets of stimuli that were identical in terms of their basic features, but differed in the combinations of these features. The logic of these analyses was similar to those implemented in previous work of colour and motion [18], and colour and form [19]. Importantly, above-chance decoding in these analyses indicates evidence of feature conjunction coding for a given pair of features. We found reliable decoding of most conjunctions, indicating a transitory stage from individual feature coding to processing higher order visual attributes (e.g., texture, shape). Conjunction coding was typically later than individual feature processing, indicating distinct neural populations from those involved in individual feature processing. Even when taking into account that lower overall classification can lead to later calculated onsets [53], the observed dynamics and

peaks for the conjunction analyses looked notably different from the single feature analyses, supporting different mechanisms and neural populations for coding conjunctions than for individual feature processing. Interestingly, previous work has found colour conjunction coding in V1-V3 [18,19], which was interpreted as "early" feature conjunctions, yet here we find that conjunction coding occurred later than individual feature coding. This delay points to an important role for recurrence and feedback within early visual regions for perception.

The dynamics of conjunction decoding can give insight into the processes underlying feature coding. Conjunction coding was shorter than that for individual features, which could signify a transitory phase to processing conjunctions of more than two features. Crucially, individual feature coding was maintained during conjunction coding, indicating that ongoing single feature processing might be necessary for computations of conjunctions between features. Different dynamics across different 2-way conjunctions suggest different mechanisms for binding different features. These findings support previous research demonstrating a transition from feature to conjunction coding throughout the visual cortex [20]. Yet, the maintained feature coding suggests that there is not a clear transformation in processing from feature coding to conjunction coding, as previously thought, and rather suggests that there is a complex interplay between recurrent and feedback processes within different regions specialised for different mechanisms. Although conjunctions of more than two features were too complex to investigate here, these results lay the foundation for future work into more complex feature conjunction coding in the brain.

## From features to percepts

An important missing piece of the visual perception puzzle is the extent to which the neural coding of basic visual features contribute to behaviour. Unpacking this relationship allows us to understand how feature processing leads to perception. Our large 4-dimensional stimulus set presents an interesting challenge for assessing behaviour. As an overall judgement of perception, we had participants make "odd-one-out" choices for three simultaneously presented images, allowing choices to be made on any of the four features in each stimulus or some combination of features. Previous work has used the odd-one-out triplet task to great effect in finding the dimensions underlying similarities in a large stimulus set [68,69]. In our case the underlying dimensions were known, which allows us to estimate the weight people give to each dimension. Interestingly, we found that all four features contributed to behaviour to varying levels. Stimulus orientation, contrast, spatial frequency, and colour models each correlated with behavioural similarity ratings, indicating that behaviour was based on a complex weighting of the different features by which the stimuli varied. One interesting finding in the behaviour is that orientation seemed to be the poorest predictor of perceptual similarity, echoing poorer classification for orientation relative to the other features. Spatial frequency and colour accounted for the most variance in behaviour. The emergence of a multi-dimensional space using such a simple behavioural measure supports the validity of the triplet approach for higher level vision [68], and shows how powerful a simple approach can be.

The richness of our behavioural results was also evident by its relationship to measured neural responses evoked by the same stimuli. Previous work that made similar comparisons between neural data and behavioural data from separate participants has shown that perceptual similarity correlates with neural responses [e.g., 70,71], even to the point that behaviour can explain the most variance in neural responses [6,23,70,72,73], indicating that perceptual decision-making might be based on a direct behavioural read out of the patterns of stimulus-based neural responses. Here we used stimuli that are highly controlled and have little semantic meaning, which allowed us to carefully study the dynamics of individual feature

representations as well as nonlinear interactions between features and their relationship to overall form perception. Our approach varies from the recent push to study complex stimuli, which has yielded much insight into ecologically valid perception [26,74–76], yet our ability to control the stimulus features conveys an advantage not yet possible (or much more difficult) with real world stimuli. Our results showed that perceptual similarity yielded reliable correlations with neural responses to the stimuli from very early stages of processing, only 80ms after image presentation. Early neural responses were therefore highly reflective of behaviour. It should be noted that we only had four stimulus levels of each feature (e.g., four different spatial frequencies), and the results might be somewhat dependent on these chosen values, so future work could interrogate this feature space further. Yet, the reliable early correlations are particularly notable given our large stimulus set, which meant that rank-order correlations between neural and perceptual similarity were performed on 32640 pairwise dissimilarity values. Further, our stimuli all had identical retinotopic projections, meaning large position-based neural differences could not contribute to the neural-perceptual relationship as it could in previous work [6,70,72]. One open question that remains is whether the neural-behaviour relationship varies according to how much the stimuli are reliant on semantic processes. Overall, early and prolonged correlations between neural patterns of activity and perceptual judgements points to the basic features of orientation, colour, spatial frequency, and contrast being primary contributors to the perceived form of our simple experimental stimuli.

## Methods

Raw and preprocessed data are available online through openneuro: https://openneuro.org/datasets/ds004357. Supplementary Material and analysis scripts are available on github: https://github.com/Tijl/features-eeg.

### Ethics statement

This study was approved by the University of Sydney ethics committee (project number: 2016/849) and informed consent was obtained from all participants. Written consent was obtained from all participants.

### Participants

There were 16 participants (11 female, 5 male, age range 18–27 years) for the electroencephalography (EEG) experiment recruited through the University of Sydney in return for course credit or payment. There were 530 participants for the online behavioural experiment, recruited via SONA in return for course credit.

### Stimuli

Stimuli were 256 oriented gratings that varied across four feature dimensions: colour, spatial frequency, contrast, and orientation (Fig 1A). The stimuli were 256 × 256 pixel sinusoidal patches with a circular mask constructed using GratingStim in PsychoPy [77] with specified orientations, colours, spatial frequencies and contrast. There were four levels of each feature dimension. The four colours were RGB values [66, 10, 104; 147, 38, 103; 221, 81, 58; 252, 165, 10], approximately equidistant in colour space, and each was presented with their complementary colour as implemented in PsychoPy. Orientations were 22.5, 67.5, 112.5 and 157.5 degrees, circularly spaced. Spatial frequencies were 2.17, 1.58, 0.98, 0.39 cycles per degree, and contrast was .9, .7, .5 and .3, both linearly spaced. The experimental stimuli consisted of all 256 possible combinations of the features (4 orientations × 4 spatial frequencies × 4 colours × 4 contrast

levels). Note that the levels of each feature were designed to be approximately equidistant in stimulus space, but these do not necessarily reflect equidistant neural or perceptual responses; our focus was to include a range of stimuli that varied along four feature dimensions: equally spaced sampling of the dimensions was not crucial to estimate featural dynamics. To avoid phase-related effects, the phase of the stimulus gratings varied randomly on each presentation in the EEG experiment. For the behavioural experiment, a specific (but randomly chosen) phase was used for each stimulus. Phases used in the behavioural experiment are shown in Fig 1.

### Online behavioural experiment

Behavioural similarity judgements were collected using an online behavioural experiment [78], programmed in jsPsych [79] and hosted using Jatos [80]. The experiment involved the display of three stimuli, and participants were asked to select the odd one out. This design has been used previously to yield high quality similarity spaces [68]. In total, 530 participants performed 500 trials each, yielding a total of 265,000 similarity judgements. To create a behavioural dissimilarity matrix, the similarity of a pair of stimuli was computed as the number of trials where the pair was presented together but not chosen as the odd one out, divided by the total number of trials that included the pair. The similarity scores were then converted into dissimilarity scores by taking 1 minus the similarity scores.

### EEG experiment

Neural responses were collected using electroencephalography (EEG) while participants viewed the experimental stimuli and performed an orthogonal task (Fig 1B). Participants viewed the stimuli on a 60 Hz ASUS VG236H 1920 x 1080 pixel gaming monitor from approximately 57cm away in a dimly lit room (exact luminance values were not recorded). Images were presented centrally at $6.5 \times 6.5$ degrees of visual angle. They were each presented for 16.67 ms duration, in sequences at 6.67Hz (133.33 ms ISI) and 20Hz (33.33 ms ISI; Fig 1B). Each sequence consisted of all 256 stimuli presented in random order. An event trigger was sent over the parallel port at every stimulus onset event. There were 80 sequences across the whole experiment, with each stimulus presented 40 times per frequency condition. Participants were asked to fixate on a black bullseye which was presented one second before the sequence began and superimposed on top of all stimuli. The task was to detect when the fixation bullseye (presented centrally at 0.6 degrees visual angle) changed to a filled circle target and respond using a button press. Targets occurred 2–4 times per sequence.

### EEG recording and preprocessing

Continuous neural data were collected using a 128-channel BrainVision ActiChamp EEG system using the international five percent system for electrode placement over the scalp [81]. Data were online referenced to FCz and digitised at 1000 Hz. All preprocessing was performed offline using EEGLAB [82]. Basic processing was performed: data were filtered using 0.1 Hz high pass and 100 Hz low pass filters, using EEGlab default Hamming windowed sinc FIR zero-phase filters. Epochs were constructed from 100ms prior to 600ms after each image presentation and downsampled to 250 Hz. No electrode interpolation or artefact rejection was performed [cf. 83]. We did not control for eye movements or other artefacts in the analyses because multivariate classification is robust to random noise and artefacts [84]. Furthermore, our design minimises eye movements because all images were presented at fixation, for short durations and in random order.

## Analyses

**Stimulus feature decoding.**   Time-resolved multivariate pattern (decoding) analyses [53] were performed to determine the temporal dynamics of visual feature coding in the brain. We decoded the EEG channel voltages across the 128 electrodes using CoSMoMVPA [85] in MATLAB. For each time point, data were pooled across all electrodes, and we tested whether a classifier could discriminate between patterns of neural activity in response to different stimulus orientations, spatial frequencies, colours, and contrast levels. Regularised linear discriminant analysis (LDA) classifiers were used to decode neural responses to the four levels of each stimulus feature (e.g., stimulus contrast 0.9 vs 0.7 vs 0.5 vs 0.3). Classification was performed separately for each feature, time point, and image presentation rate. A leave-one-sequence-out cross-validation approach was used, where the classifier was trained on trials from all but one sequence (39) and tested on the left-out sequence (1), and this was repeated 40 times so each sequence was the test set once. Each sequence contained 256 stimuli, with a quarter of these belonging to each stimulus feature level. Thus, each feature analysis contained the same trials, but in different combinations. Decoding accuracy was calculated as the proportion of correct cross-validated classifier predictions and compared to chance performance of 0.25. To obtain an estimation of spatial contributions to decoding performance, we performed a channel-searchlight analysis. For each EEG channel, we selected the nearest 4 neighbouring channels and performed the same decoding analysis as above, storing the accuracy at the centre channel, obtaining a time-varying spatial map of cross-validated decoding accuracies for each participant. Finally, we performed temporal generalisation [37] to investigate how the decodable signals change over time. For this analysis, we performed the same decoding approach as above, but for each time point, we tested the classifier on all time points in the epoch. This results in a time-time matrix of cross-validated decoding accuracies for each participant.

**Feature interactions.**   To assess how coding of different features interact, we performed the same time-resolved decoding analyses using subsets of the data corresponding with specific stimulus features. We classified each of the four stimulus features for each level of the remaining three features; for example, decoding stimulus orientation for each level of contrast, spatial frequency, and colour. The same decoding scheme (i.e., 4 class, leave-one-sequence-out cross-validation) was used as in the original feature decoding analysis, but with only one quarter of the trials. Thus, chance level was still 0.25. To compare across conditions, we assessed peak decoding accuracy during a 20ms time window centred on the peak of the original feature analysis (i.e., 120ms for orientation, 112ms for SF, 112ms for colour and 96ms for contrast). The mean accuracy was calculated for each time window per participant and statistically compared across levels of a given feature (Fig 4).

**Feature conjunctions.**   To assess the conjunctions of pairs of features (e.g., SF × orientation), we decoded groups of stimuli that each had the same absolute features, but varied in their feature *combinations* (see Fig 5). For example, low spatial frequency high contrast stimuli and high spatial frequency low contrast stimuli (class 1) versus low spatial frequency low contrast and high spatial frequency high contrast stimuli (class 2). For each pair of features, classification was performed for the 36 different possible stimulus combinations (e.g., SF3ori3/SF2ori2 vs SF3ori2/SF2ori3, SF3ori3/SF2ori4 vs SF3ori4/SF2ori3, etc) and then the mean was calculated. Above chance classification for this analysis reflects neural responses to the combinations of different pairs of features, indicating the coding of feature *conjunctions*. Note that this analysis requires a classifier to pick up on features that reflect differences between a group of two stimuli versus another two stimuli, but this does not mean that it is necessarily using features that are common within each group; instead, we expect that a classifier learns neural patterns associated with each of the four stimuli and can find a boundary

that separates two stimuli from two others. This analysis is very similar to previous analyses on colour-form conjunctions [19].

**Representational similarity analysis.**    To assess relationships between the neural, behavioural and feature models, we used the Representational Similarity Analysis framework [38]. Neural representational dissimilarity matrices were constructed using pairwise decoding for all pair combinations of the stimuli, resulting in a symmetrical $256 \times 256$ matrix for each time point per participant. Model dissimilarity matrices were constructed for each of the four visual features (contrast, colour, spatial frequency, and orientation) based on the rank similarity of that feature in the stimulus set (Fig 6A). For the contrast model, for example, stimuli that were presented at contrast level 0.9 were most similar to level 0.7 (dissimilarity = 1), more dissimilar to level 0.5 (dissimilarity = 2) and most dissimilar to level 0.3 (dissimilarity = 3). The same scheme was used for spatial frequency and colour. Due to the circular nature of orientation, the orientation model was slightly different; stimuli orientated at 22.5 degrees were as similar to 67.5 degrees (45 degree difference; dissimilarity = 1) as they were to 157.5(-22.5) degrees (45 degrees difference; dissimilarity = 1). Importantly, each of the four feature models were orthogonal (orientation versus other models $r$ = -.008, $p$ = .171; the others $r$ = -.007, $p$ = .189). Finally, behavioural dissimilarity matrices were constructed from the results of the odd-one-out task, where higher dissimilarity indicated pairs of stimuli where one of the stimuli tended to be chosen as the odd-one-out when they were presented together (Fig 6B).

To assess how visual features of the stimuli relate to behavioural responses of their similarity, we correlated the behavioural results with each of the four feature models. Then, to assess how stimulus features and behavioural judgements are related to the neural responses of the stimuli, we correlated the neural dissimilarity matrices with the four feature models and the behavioural model, separately for each time point and participant. Each comparison was performed using Spearman correlation for the unique values in the dissimilarity matrices (i.e., the lower triangles, 32640 values), yielding a very robust estimate of the relationship between each model.

## Statistical inference

The decoding analyses were performed at the individual subject level, and the group-level decoding accuracies or correlations were statistically compared to chance level. We used Bayes factors [86–88], as implemented in the Bayesfactor R package [89] and its corresponding implementation for time-series neuroimaging data [88]. The prior for the null-hypothesis was set at chance level (for decoding) or zero (for the correlations). The alternative hypothesis prior was an interval ranging from small effect sizes to infinity, accounting for small above-chance results as a result of noise [86,88]. For calculating the onset of above chance decoding or correlations, we found the first three consecutive time points where all Bayes Factors were >10. To be able to compare onset and peak times, we calculated 95% confidence intervals by using a leave-two-participants-out jackknifing approach, where we calculated the onset and peak for all possible leave-two-out permutations (n = 120 permutations for 16 participants), and took the 95th percentile of the resulting distributions.

## Supporting information

**S1 Fig. Sensor searchlight decoding of each feature across time, 6.67 Hz condition (window size: 5 channels).**
(TIFF)

**S2 Fig. Sensor searchlight decoding of each feature across time, 20 Hz condition (window size: 5 channels).**
(TIFF)

**S3 Fig. Time generalisation results for all conditions.**
(TIFF)

**S4 Fig. Decoding one feature for every level of another feature for 6.67 Hz condition.** Plots on the diagonal show overall decoding for that feature when including all trials.
(TIFF)

**S5 Fig. Decoding one feature for every level of another feature for 20 Hz condition.** Plots on the diagonal show overall decoding for that feature when including all trials.
(TIFF)

**S6 Fig. Peak decoding of one feature for every level of another feature for 20Hz condition.**
(TIFF)

**S7 Fig. Feature conjunction analyses for 6.67Hz condition, plotted separately for each of the 36 contrasts performed per feature combination.** Within each plot, it is evident that the different contrasts varied in decoding performance, but were consistent in terms of the dynamics (e.g., peaks). Onsets and peaks of individual features and mean conjunctions are plotted with 95% confidence intervals; onsets are below the chance level and peaks are above. Note the different y-axis scales per row.
(TIFF)

**S8 Fig. Feature conjunction analyses for 20Hz condition, plotted separately for each of the 36 contrasts performed per feature combination.** Within each plot, it is evident that the different contrasts varied in decoding performance, but were consistent in terms of the dynamics (e.g., peaks). Onsets and peaks of individual features and mean conjunctions are plotted with 95% confidence intervals; onsets are below the chance level and peaks are above. Note the different y-axis scales per row.
(TIFF)

## Acknowledgments

The authors acknowledge the Sydney Informatics Hub and the University of Sydney's high performance computing cluster Artemis for providing the high-performance computing resources that contributed to these research results.

## Author Contributions

**Conceptualization:** Tijl Grootswagers, Amanda K. Robinson.

**Data curation:** Tijl Grootswagers.

**Formal analysis:** Tijl Grootswagers, Amanda K. Robinson, Sophia M. Shatek.

**Funding acquisition:** Amanda K. Robinson, Thomas A. Carlson.

**Investigation:** Tijl Grootswagers, Amanda K. Robinson, Sophia M. Shatek.

**Methodology:** Tijl Grootswagers, Amanda K. Robinson.

**Visualization:** Tijl Grootswagers, Amanda K. Robinson.

**Writing – original draft:** Tijl Grootswagers, Amanda K. Robinson.

**Writing – review & editing:** Tijl Grootswagers, Amanda K. Robinson, Sophia M. Shatek, Thomas A. Carlson.

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
