## [Decision Letter · Decision Letter 0]

20 Jul 2023

Dear Dr Grootswagers,

First of all my apologies for getting back so late. It sometimes happen that a paper falls in the long tail due to a combination of unavailabilities of editors and reviewers.

Thank you very much for submitting your manuscript "Mapping the Dynamics of Visual Feature Coding: Insights into Perception and Integration" for consideration at PLOS Computational Biology.

As with all papers reviewed by the journal, your manuscript was reviewed by members of the editorial board and by several independent reviewers. In light of the reviews (below this email), we would like to invite the resubmission of a significantly-revised version that takes into account the reviewers' comments. Particular attention should be paid to insert the work in the current state of the art and on defining the original contribution.

We cannot make any decision about publication until we have seen the revised manuscript and your response to the reviewers' comments. Your revised manuscript is also likely to be sent to reviewers for further evaluation.

Sincerely,

Daniele Marinazzo

Section Editor

PLOS Computational Biology

Daniele Marinazzo

Section Editor

PLOS Computational Biology

Reviewer's Responses to Questions

**Comments to the Authors:**

Reviewer #1: Review is uploaded as an attachement

Reviewer #2: In this study, the authors apply neural decoding to evoked potentials (measured via EEG) while human participants (n = 16) viewed Gabor stimuli that systematically varied in contrast, orientation, spatial frequency, and color while engaged in a distractor task at fixation. While the authors explore the decodability of those Gabor features by themselves (replicating previous work in the temporal domain), their main focus was on the dynamics of the decoding results based on the interactions and conjunctions of those features. They also relate their decoding results to the behavioral performance of an independent group of participants engaged in a simple ‘odd-one-out’ task using the same stimuli. The results confirmed previous work (differences in EEG and MEG signal sources aside) and revealed interesting decoding results as a function of different feature conjunctions that may support earlier results from behavioral psychophysics. The also show that the backward masking inherent in their EEG stimulus paradigm seemed to have little impact on feature decodability. Overall, I found the manuscript to be clearly written and relatively straightforward to follow. I do have some questions/concerns for the authors to consider (given below in no particular order).

1) The question of how ‘low-level’ feature encoding contributes to task performance (or perceptual experience) in humans is not new. And, despite the authors’ claim, a significant amount has been learned about this question from paradigms ranging from EEG, MEG, to fMRI. The authors paint a picture that leads the reader to believe that we know very little in the way of an answer to that question – there are many in the visual neuroscience community that would strongly disagree (at least in terms of how the study is motivated in the introduction). The authors would be well advised to recast their motivation for the current study in light of the recent (and semi-recent) EEG work from the labs of Michelle Greene, Iris Groen, Bruce Hansen, Assaf Harel, and Steven Scholte. All have contributed insight into the temporal dynamics of filter based features (such as those explored in the current study) in support of scene perception and related task performance -- all with stimuli and tasks that are much more ecologically valid than what is employed here. Related to that, the authors note that human neuroimaging has mainly focused on low-level perceptual psychophysics, which is incorrect (see recent work by those, and others, noted above). In fact, one could easily classify the psychophysical paradigm employed in the current study as a ‘low-level’ psychophysical task.

2) Mapping the temporal dynamics of simple visual features when simultaneously presented (e.g., a color Gabor of a certain orientation, spatial frequency, and contrast) is an important undertaking. However, I can’t help but wonder how much of any of what is reported here would hold if the stimuli were more ecologically valid. Our visual world is filled with structures that simultaneously activate a broad range of visual neurons that vary in their feature tuning. Decades of studies have shown that those neurons interact – resulting in myriad nonlinear competitions for which population will contribute to the percept and/or support a given task. This is why, at approximately the turn of the century, many in the field shifted away from simple Gabor stimuli in favor of more complex stimuli – then seeking models composed of various filters to decompose those stimuli in a way that maps onto neural or behavioral response variance. The linkage to behavior in the current study is nice, but again, the task itself is just a discrimination task between Gabors. All that said, I’m not saying that the current study has nothing to offer, just that great caution should be taken (explicitly stated) in terms of just how much of what is reported here will hold when it comes to how the system actually works in the real-world.

3) Related to the above, the 1980s (and late 1970s for that matter) are filled with psychophysical studies that used Gabors (or sinusoidal gratings) composed of systematically varying features – all attempting to elucidate how such features interact (the primary focus of the current study)….far too many to list here, but it would at least be nice to know how much of what is reported here supports some of that earlier work.

4) Please provide more detail concerning the group-level analyses – exactly how were the data from each participant entered into the analysis (give an appendix if it’s already made clear in an earlier publication by this group). It doesn’t seem like the decoding was run on an individual-by-individual participant basis – the confidence intervals are strikingly narrow, suggesting that it wasn’t). If not, why? It would be very informative to the audience to get a good sense of the extent to which the reported results are observable at the individual level. Similarly, it would be good to report signal quality tests on each participant (e.g., split-half prediction or something of that ilk). The differences in cortical folding can lead to certain sources being obscured on the scalp and add unnecessary noise to the analyses. No mention of signal quality checking is given here.

5) The EEG task presents Gabors to the fovea (extending only a degree or so into the periphery), so why such a low range of spatial frequencies (2.17cpd is the highest spatial frequency used here). Similarly, why did the authors choose to leave out the cardinal orientations? Might the odd reduction in decoding accuracy in color with decreasing spatial frequency be accounted for by this somehow? The color system is largely lowpass, so I found that result more confusing than interesting.

6) Why are there two different SOA paradigms used in the current study? Some justification (or at least some rationale) should be given in the manuscript. Also, how large was the bullseye for the distractor task in the EEG experiment?

7) Crucial details about the monitor have been left out (outside of stating that it has a temporal resolution of 60Hz). Please provide the specs on the monitor. Also, I assume that the stimuli were linearized – please mention that. If it wasn’t, then the results will need to be considered in the context of the stimuli not being linear with respect to luminance.

8) For the EEG data, why were they down sampled to 250Hz? Also, given the lowpass filter had a cut off at 100Hz, I assume that there was no electrical noise? If a notch filter was used, it should be reported.

9) Given that participants in the EEG experiment and those in the psychophysical experiment were doing different tasks, I don’t think it’s appropriate to make the connection between poor classification for orientation and poor predictability of perceptual similarity. The behavioral participants clearly prioritized some features of others in the triplet tsk – the same cannot be said for those in the EEG experiment doing a task that didn’t require them to use the Gabor features at all.

Reviewer #3: In this manuscript, the authors use an EEG paradigm to trace the neural dynamics of visual features and their interactions along visual hierarchy, using the Gabor stimuli displayed in a rapid sequence (6.67Hz vs. 20Hz) that each stimulus parametrically varies in feature combinations along 4 dimensions (orientation, spatial frequency, color, and contrast). Based on a multivariate decoding analysis, they then study how single feature and their non-linear conjunction are dynamic represented in visual cortex. Additionally, they try to explore the behavioral relevance of these neural signatures using representational similarity analysis (RSA), in which the behavioral matrix (perceptual similarity between the tested Gabors) is calculated using a separately data set tested on a new group of participants.

Overall, the general approach taken in this paper – applying different classifiers to the same set of parameterized stimuli – is promising to study the neural dynamics of visual information processing in the brain. The question the authors set out to investigate is relevant and the manuscript is well written. However, I have multiple comments listed below. To preview, my main concern is about the limited knowledge that this paper can add to the field. There is little novel insight that change or add to our understanding of the Information processing dynamics and computations along the visual hierarchy.

Major comments:

1. The authors trace the coding dynamics of each feature (orientation, SF, colour, and contrast) in the brain and show a temporal profile displaying two representational peaks, one around 100ms and the other 170m. Such typical effects have been observed in so many EEG/MEG studies involving visual recognition, categorization or with just passive viewing paradigm, and at both object/image level and at decomposed single feature level. Additionally, it has been well demonstrated that multiple features of a given stimulus are represented along the visual hierarchy in parallel, with the early peak (~100ms) activates the early visual cortex (same as the reported results in this manuscript, see Figure 2 topological map) and the later peak (~170) propagates towards the occipital-temporal junction (the authors did not check this in their manuscript). Though the authors state that their analysis shows different coding dynamics of each feature in terms of onset/peak time and temporal generalization pattern based on the group average analysis (in Figure 2), I quite doubt the reliability of such results due to the very high individual difference in single feature representations in the brain found in previous and my own MEG studies. This issue can be probably resolved by adding individual-level analysis together with prevalence analysis, even so this paper only shows a phenomenon rather than provides explanation of underlying mechanism. And we still don’t know why our brain represents these features differently, what is their perceptual or cognitive significance? Taken together, the feature-level dynamics in early visual cortex revealed in current study can be well predicted as which has been well-investigated in previous studies.

2. The authors find interesting asymmetric interaction patterns between the coding of different features. However, I have problem of understanding these results (Figure 3). How such representational interactions emerge during the passive viewing task? Why our brain displays such asymmetric feature interactions? To answer the first question, the authors should locate where and when such interaction happens in the brain at least at channel level (or better at source level using source reconstruction analysis), which may provide some evidence to promote our understanding. However, for the later question in above, I don’t think the authors can provide good answer based on their current design. What I wonder is whether the observed results constitute a general property of perceptual processing (e.g., generalize to other stimuli type that can be decomposed into similar feature dimensions, and to tasks with different cognition demands such as passive viewing, recognition, categorization) or are very task-specific (e.g., constrained by the paradigm used in current study). Even it was the later, we still need to have a mechanism explanation over the observed phenomenon (as I raised in comment 1). However, based on their current experimental manipulations (the 6.67Hz vs. 20Hz stimuli display rate), I don’t think it targets to understand the nature of the feature interactions. So again, the purely interaction phenomenon revealed in current results add very limited knowledge to the related research fields.

3. The authors designed a very clever classifier to investigate the feature conjunctions in the brain. Based on my understanding, the classifier is an XOR solution so that it requires a (non-linear) integration of 2 features together to generate an output. I quite appreciate this idea. However, tracing low- (and high-) level feature integration dynamics is not new in human EEG/MEG studies (see K Jaworska et al., 2022, elife, with a similar pattern but with much better spatial-temporal resolutions) which limits my enthusiasm for this result. Also, as the analysis approach mixes many different components, I found it is very difficult to interpretate and evaluate their conjunction results. For example, for each feature-pair conjunction, it averages across 36 combination possibilities; however, due to a different feature-by-feature interaction pattern, I would expect different temporal profile (e.g., onset/peak/amplitude) of each combination possibility, and then averaging them out can lead to some biased conclusion. Additionally, the sustained conjunction decoding curves and their multiple peaks (see Figure 4B) suggest different brain sources (e.g., brain regions/voxels) contributing to the feature integration at different processing stages; however, the authors mix everything up and then the results become very uninformative to me. Taken together, I doubt the reliability of this conjunction results. For example, the authors show a counter-intuitive result to me, i.e., early feature conjunction before the individual feature representation (see Figure 4B, colour*contract and SF*contrast panel).

4. The current design does not have experimental power to investigation the behavioural relevance of feature coding along visual hierarchy, for 4 reasons: 1) they across-corelate the data obtained in 2 independent experiments, that 2) recruit different participants to 3) perform very different tasks (one is passive viewing with an orthogonal stimuli-irrelevant task, and the other is a stimulus-based similarity judgement that requires strong feature processing demand) in 4) different experimental environments (online vs. in scanner) Due to the high individual variability in their ability of Garbor-related feature processing and the significant manipulation of task demand on both the early low-level feature processing in occipital regions and late high-level feature integration in ventral temporal regions (together with frontal modulation), I don’t think the reported data cannot support the behavioural-relevance claims made in this paper.

5. An important manipulation in this study is using RSVP paradigm to control the displaying rate of the visual stimuli in 2 conditions (6.67Hz vs. 20Hz), however the authors provide no explanation why they introduce such manipulation. In their introduction, they should include scientific motivation of such design with appropriate cited references, to explain why they choose these two conditions not others, and what kind of hypothesis they can test based on such Hz (i.e., different ISI-SOA combination) settings. Without this information, it’s very difficult to interpretate and evaluate their null effects reported in current study.

Minor comments:

1. There are 4 levels of each feature dimension. The authors should explain how they choose the feature range with 4 equally sampled level, e.g., based on parameters used in previous studies or based on a psychophysical piloting? I notice they did not perceptually calibrate the feature levels per participant, which exacerbates my concern about the potentially large individual differences (related to my major concerns raised in point 1 and 4).

2. The authors use “perceptual experience” in their abstract and throughout the manuscript. It’s not clear to me what the *perceptual experience* means in their work. I believe researchers in the field of object categorization vs. perceptual learning will give very different explanation for this term.

**Have the authors made all data and (if applicable) computational code underlying the findings in their manuscript fully available?**

Reviewer #1: Yes

Reviewer #2: Yes

Reviewer #3: None

PLOS authors have the option to publish the peer review history of their article (what does this mean?). If published, this will include your full peer review and any attached files.

Reviewer #1: **Yes: **Benedikt Ehinger

Reviewer #2: No

Reviewer #3: No
---

## [Decision Letter · Decision Letter 1]

13 Dec 2023

Dear Dr Grootswagers,

We are pleased to inform you that your manuscript 'Mapping the Dynamics of Visual Feature Coding: Insights into Perception and Integration' has been provisionally accepted for publication in PLOS Computational Biology.

Best regards,

Daniele Marinazzo

Section Editor

PLOS Computational Biology

Daniele Marinazzo

Section Editor

PLOS Computational Biology

Reviewer's Responses to Questions

**Comments to the Authors: **

Reviewer #2: The authors have carefully considered and thoroughly addressed my concerns. Thank you for that. The only exception, which appears to be a difference in opinion, is the authors' choice to continue to speculate on a comparison between neural data obtained from one task and independent behavioral data from another task. I know it is common to compare neural data from one sample to behavior obtained from an independent sample. I do not take issue with that (I've done it myself many times). However, the tasks should match. That's just good science. And, Greene and Hansen 2020 (which the authors chose to cite) actually show that task demands can dictate differential feature use (even the 'low level' features), so I'm not convinced that the mechanisms studied here simply provide a readout regardless of task. At any rate, I don't think this debate can be resolved here and do not expect the authors to reply to this comment nor am I asking them to modify their manuscript unless they choose to do so (I've already recommended that the manuscript be accepted in its revised form). I would, however, advise the authors' to plan their future work so that the behavioral data can be either taken directly from the recording sessions (which I know isn't always possible), or that the tasks match if gathering behavioral data from an independent sample. Unless, of course, the goal is to argue against task demands playing a role in shaping the integration of feature processing (which was *not* the aim of the current study).

**Have the authors made all data and (if applicable) computational code underlying the findings in their manuscript fully available?**

Reviewer #2: Yes

PLOS authors have the option to publish the peer review history of their article (what does this mean?). If published, this will include your full peer review and any attached files.

Reviewer #2: No

---

## [Editor Report · Acceptance letter]

2 Jan 2024

PCOMPBIOL-D-23-00790R1 

Mapping the Dynamics of Visual Feature Coding: Insights into Perception and Integration

Dear Dr Grootswagers,

I am pleased to inform you that your manuscript has been formally accepted for publication in PLOS Computational Biology. Your manuscript is now with our production department and you will be notified of the publication date in due course.

With kind regards,

Lilla Horvath
